# Improved visual detection of DNA amplification using pyridylazophenol metal sensing dyes

Yinhua Zhang[1], Eric A. Hunt [1], Esta Tamanaha[1], Ivan R. Corrêa Jr. [1] & Nathan A. Tanner [1✉]

Detection of nucleic acid amplification has typically required sophisticated laboratory instrumentation, but as the amplification techniques have moved away from the lab, complementary detection techniques have been implemented to facilitate point-of-care, field, and even at-home applications. Simple visual detection approaches have been widely used for isothermal amplification methods, but have generally displayed weak color changes or been highly sensitive to sample and atmospheric effects. Here we describe the use of pyridylazophenol dyes and binding to manganese ion to produce a strong visible color that changes in response to nucleic acid amplification. This detection approach is easily quantitated with absorbance, rapidly and clearly visible by eye, robust to sample effects, and notably compatible with both isothermal and PCR amplification. Nucleic acid amplification and molecular diagnostic methods are being used in an increasing number of novel applications and settings, and the ability to reliably and sensitively detect them without the need for additional instrumentation will enable even more access to these powerful techniques.

[1] New England Biolabs, Ipswich, MA, USA. ✉email: tanner@neb.com

Nucleic acid amplification is a standard laboratory practice with varied approaches and applications from molecular biology to clinical diagnostics. Regardless of method or use, the amplification reaction must be detected in an accessible manner. Analysis of the product by gel electrophoresis has long been the gold standard technique, as it provides information about product size, yield and specificity. However, electrophoresis is laborious, not suited to use outside a laboratory, only semi-quantitative, and introduces a risk of laboratory space and equipment contamination during sample transfer. Numerous fluorescence detection methods amicable to high-throughput analysis were developed along with the introduction of real-time quantitative PCR (qPCR)[1], primarily double-stranded DNA (dsDNA) binding dyes (e.g., SYBR Green) and sequence-specific fluorogenic probes such as TaqMan hydrolysis probes. More recently the use of isothermal amplification techniques such as loop-mediated isothermal amplification (LAMP)[2] has attracted significant attention due to lower instrumentation demands compared to the thermocycling requirement of PCR. Many LAMP detection strategies utilize the same fluorescence principles as PCR including dsDNA binding dyes (e.g., SYTO-9) and probe-based detection methods (e.g., DARQ[3], QUASR[4], and molecular beacons[5,6]). Other isothermal amplification methods, such as SDA[7,8], RPA[9], NASBA/TMA[10] require probe-based detection because they are prone to non-specific amplification. Detection of fluorescence in any of these approaches requires relatively sophisticated instrumentation, similar to real-time qPCR machines, increasing the cost and complexity of an otherwise simplified isothermal amplification scheme.

An additional benefit of LAMP is the ability to utilize visual detection methods based on a change in color or turbidity of the reaction. Amplification by LAMP generates 5- to 10-fold more DNA product than a typical PCR, and that large DNA yield, or the byproducts of its synthesis, can be exploited to simplify the reaction detection. For example, the DNA-sensing dye malachite green produces a slight visual color change from dark blue to light blue upon positive LAMP reactions[11,12]. An example byproduct is the accumulated inorganic pyrophosphate (PPi), which is generated from every nucleotide addition during DNA synthesis and may reach a concentration high enough to form a precipitate with $Mg^{2+}$, leading to a turbid appearance in the reaction[13]. In practice, turbidity may be difficult to discern by the naked eye and is more reliably measured using appropriate instrumentation. PPi precipitation with $Mg^{2+}$ also leads to a drop in soluble $Mg^{2+}$ concentration that can be sensed by metal binding colorimetric dyes, such as hydroxynaphthol blue (HNB)[14] and eriochrome black T (EBT)[15,16], or the metal sensing dyes, such as calcein[17], whose detection can be enhanced with UV irradiation.

The clearest visual detection of LAMP to date has come from the use of pH sensitive dyes, whose high color contrast enables discrimination of positive from negative reactions[18]. In this approach, LAMP is performed in a weakly buffered solution and the acidification of the reaction during DNA polymerization drives the color change of the pH sensitive dye. This method is particularly suitable for point-of-care settings and has been used in the detection of human pathogens[19,20], field surveillance of mosquitos[21] and plant viral infections[22–24]. During the ongoing COVID-19 pandemic, pH-based LAMP detection has been applied to the detection of SARS-CoV-2 in numerous clinical diagnostics[25–33], resulting in the first ever molecular test approved for at-home use[34].

However, colorimetric detection based on pH change has some inherent constraints. pH-based detection is limited to samples without significant pH variation and precludes the use of stabilizing buffering solutions[25,35]. In addition, this approach is difficult to adapt for monitoring PCR and other amplification methods[18].

Toward the goal of achieving high-contrast but pH-independent visual detection, we screened a range of metal-sensing dyes for their ability to provide robust color change upon DNA amplification in fully buffered reactions. We describe here two related dyes, 2-(5-Bromo-2-pyridylazo)-5-[N-propyl-N-(3-sulfopropyl) amino]phenol (5-Bromo-PAPS) and 2-(5-Nitro-2-pyridylazo)-5-[N-n-propyl-N-(3-sulfopropyl)amino]phenol (5-Nitro-PAPS) that meet the necessary criteria and enable improved colorimetric LAMP. These PAPS dyes have been previously used for colorimetric detection of various metal ions[36–38] and of the amino acid homocysteine[39] with high sensitivity. We found that in the presence of $Mn^{2+}$ they displayed a remarkable color shift in response to DNA amplification, producing a high color contrast between positive and no template control (NTC) reactions, and could be easily deployed in quantitative absorbance measurements. Importantly, this colorimetric detection system worked robustly in both LAMP and PCR amplification. We believe this improved colorimetric detection system could be further expanded to other DNA amplification workflows and facilitate the development of simplified molecular diagnostic tests.

## Results and discussion
We screened for new visual reporters for LAMP amplification in fully buffered (20 mM Tris) reactions with metal binding dyes in combination with various metal ions. We observed that two pyridylazo dyes, 5-Bromo-PAPS and 5-Nitro-PAPS, displayed a variety of bright colors when complexed with certain metal ions (Supplementary Fig. 1). We found that of those, $Cu^{2+}$, $Zn^{2+}$, $Fe^{2+}$, and $Ni^{2+}$ changed 5-Bromo-PAPS and 5-Nitro-PAPS dyes to various bright colors in the initial LAMP buffer conditions, whereas $Ca^{2+}$, $Cr^{2+}$, and $Re^{3+}$ did not. However, all these complexometric metal ion-dye colors remained unchanged after LAMP amplification. $Mn^{2+}$ changed the color of 5-Bromo-PAPS from yellow to bright red when added to the reaction mixture. Strikingly, at the end of LAMP amplification, positive reactions reverted to the same yellow color as observed without $Mn^{2+}$, while NTC reactions remained an unchanged red (Fig. 1a and Supplementary Fig. 1). Similarly, 5-Nitro-PAPS changed color from yellow to dark red after adding $Mn^{2+}$ and then back to yellow upon successful LAMP amplification (Fig. 1b and Supplementary Fig. 1). A related pyridylazo dye 4-(2-pyridylazo) resorcinol (PAR) also changed color in response to LAMP amplification in the presence of $Mn^{2+}$, but the color contrast was much less pronounced: brownish yellow to yellow upon adding $Mn^{2+}$ and then back to brownish yellow after amplification (Fig. 1c). Two other dyes, HNB and EBT, which have been used as indicators for LAMP amplification based on $Mg^{2+}$ sensing, had their coloring almost completely quenched by addition of $Mn^{2+}$ (Fig. 1d, e). We also screened various other dyes that resulted in no significant change of color upon LAMP amplification, with or without $Mn^{2+}$ or other metal ions and thus focused on the PAPS dyes (Fig. 2a) for additional study.

We hypothesized that the pyridylazo dye complexes with $Mn^{2+}$ to give a red color, but the $Mn^{2+}$ is displaced by the PPi produced during the LAMP reaction, which forms an insoluble manganese(II) pyrophosphate, thereby restoring the dye's original yellow color (Fig. 2c). To demonstrate this principle, we tested the roles of key components in the proposed mechanism. First, we confirmed the importance of PPi, not the inorganic monophosphate (Pi), by adding inorganic pyrophosphatase (PPase), which converts the PPi generated by DNA polymerization to Pi. Consistent with the fact that Pi does not strongly sequester divalent metals, the addition of inorganic pyrophosphatase abrogated the red-to-yellow color change due to LAMP (Fig. 2b). Similarly, adding high concentrations of ethylenediaminetetraacetic acid (EDTA) resulted in red-to-

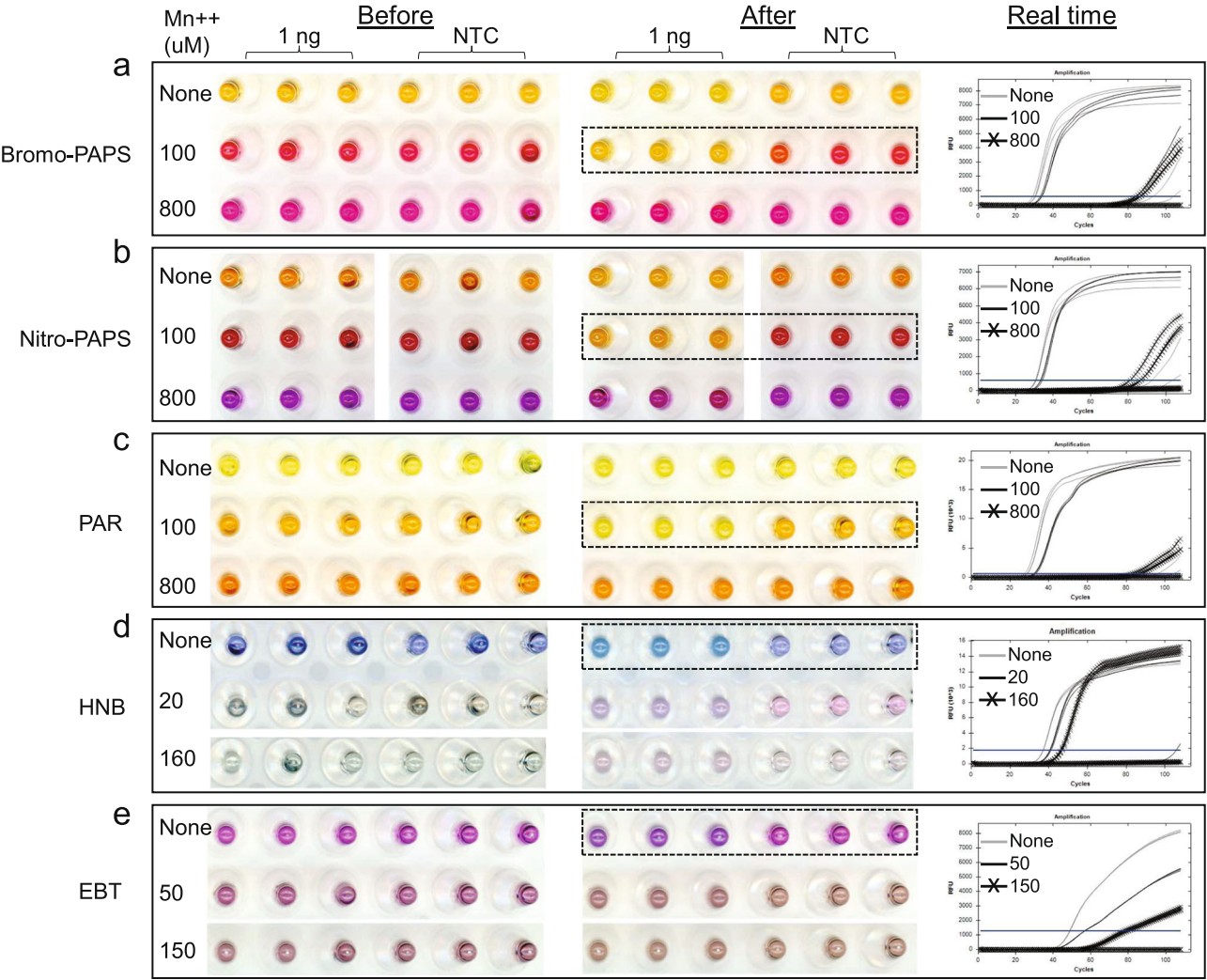

**Fig. 1 Screening metal sensing dyes as a visual reporter for LAMP amplification.** For each dye, the color of triplicate LAMP reactions is shown for before (left column) and after (middle column) the LAMP reaction with either lambda DNA (1 ng) or without (NTC). LAMP amplification was confirmed by real-time curves (right column; 1 cycle = 15"). The difference in visible dye color reduces the real time fluorescence emitted by the dsDNA binding dye, and thus the $Y$-axis for the real time curves is different due to automatic scaling by the software. Each dye was tested either without $Mn^{2+}$ or with two concentrations of $Mn^{2+}$ (100 and 800 μM). The labels for these three conditions are color-coded the same as their corresponding real-time curves. The best visual detection condition for each dye is highlighted in a dotted rectangle. **a** 75 μM 5-Bromo-PAPS. **b** 75 μM 5-Nitro-PAPS. **c** 200 μM PAR. **d** 80 μM HNB. **e** 100 μM EBT. In reactions with 800 μM $Mn^{2+}$, the amplification was significantly impaired, as shown in the real-time column of **a**–**c**. In addition, such limited amplification combined with the high concentration of $Mn^{2+}$ could not produce a color change after 40 min incubation time (middle column in **a**, **b**) likely due to insufficient production of PPi.

yellow color change immediately, independent of amplification, suggesting that EDTA chelates the $Mn^{2+}$ more strongly than PAPS.

Next, we examined optimal concentration ranges of 5-Bromo-PAPS and 5-Nitro-PAPS dyes and $Mn^{2+}$ for colorimetric detection (shown for 5-Bromo-PAPS in Fig. 3). Concentrations of each dye from 50 to 100 μM (Fig. 3a) and of $Mn^{2+}$ from 50 to 150 μM were tested. As measured by real-time fluorescence, at these concentrations, PAPS dyes did not cause any suppression of the LAMP reaction, whereas $Mn^{2+}$ caused only slight inhibition at 150 μM (Fig. 3c). All combinations provided a high-contrast visual difference between positive and NTC reactions and reliable visual detection of LAMP amplification. For subsequent studies we chose 75 μM PAPS dye with 100 μM $Mn^{2+}$ (Fig. 3b) for optimal visual contrast.

To better understand and quantitate the visual color change, we analyzed the absorbance spectrum shift of PAPS dyes in completed LAMP reactions (Fig. 4a, b). In the NTC reactions, which retained

the same red color as that of before LAMP, both PAPS dyes showed two absorbance peaks at 450 and around 550 nm. In positive reactions, the absorbance at 450 nm significantly increased while the absorption around 550 nm decreased considerably. These shifts of absorbance peaks could be used to quantitatively determine positive reactions in objective automated measurement systems such as a spectrophotometer or plate reader. Two approaches are shown in Fig. 4: using the relative absorbance difference between 450 and 550 nm (Fig. 4c) or the ratio between these peaks (Fig. 4d). Both methods provided easily distinguishable readouts of positive and negative reactions. In addition to end point analysis, the absorbance of these two peaks could be measured in real time and applied to determine amplification results, similarly to what was shown for the pH-based colorimetric method used in the high-throughput Color Health COVID-19 colorimetric LAMP assay[27]. For simple applications using smartphone cameras or basic image analysis, the processes and tools developed for colorimetric LAMP such as the

LAMP Plate Reader app (https://apps.apple.com/us/app/lamp-plate-reader/id1529271060) and hue/saturation analysis[40,41] can be used for any of the colorimetric dyes and are suitable for PAPS.

We compared the speed of visual color change relative to the dsDNA accumulation during LAMP amplification with two commercial LAMP mixes (Supplementary Fig. 2). The color change was also measured with a spectrometer and the difference between 450 and 550 nm was plotted. The results indicated that the visual color change is concurrent with the dsDNA accumulation, with visual and absorbance measurement appearing immediately following fluorescence signal from dsDNA intercalating dye, and color change visible by ~20 min incubation.

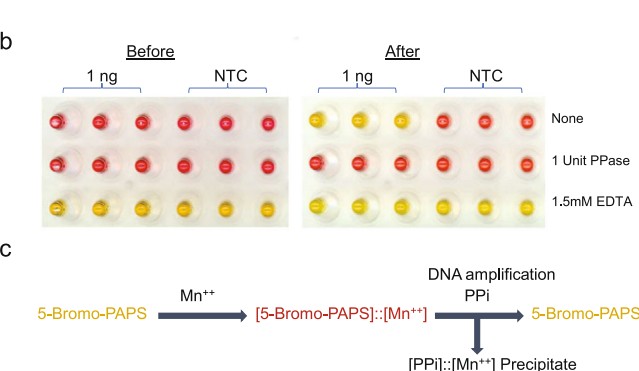

**Fig. 2 Mechanism of color change by PAPS dye in LAMP amplification. a** Structures of 5-Bromo-PAPS and 5-Nitro-PAPS. **b** Effects of degrading pyrophosphate by pyrophosphatase (PPase) or chelation of $Mn^{2+}$ by EDTA. **c** Proposed mechanism of color transitions by PAPS dyes during LAMP amplification.

As a demonstration of PAPS-based detection, we applied it for detecting synthetic SARS-CoV-2 RNA by RT-LAMP. We performed 24 reactions with input of approximately 10 copies of viral RNA per reaction, which is below the limit of detection of a commercially available kit based on a pH-dependent dye (~50 copies, SARS-CoV-2 Rapid Colorimetric LAMP Assay Kit, NEB #E2019[42,43]), in order to observe both positive and negative reactions (Fig. 5). Out of these 24 reactions, 12 showed color change as positive LAMP reaction after 40 min incubation (Fig. 5a, b). These were also determined to be positive by real-time fluorescence detection. The other 12 reactions showed no color change and matched the 8 NTC reactions which had no amplification as confirmed by real-time detection. By plotting the relative absorption change at 450 and 550 nm (Fig. 5c) or the ratio between them (Fig. 5d), values for positive reactions consistently showed a large difference from those of negative and NTC reactions, providing an unambiguous identification criterion for virus-specific amplification. We also investigated whether PAPS and $Mn^{2+}$ or both together could affect RT-LAMP detection sensitivity (Table 1). We performed tests with N2 or E1 primer sets for SARS-CoV-2 using either a single set or both sets, with or without the pH-based colorimetric LAMP stimulator guanidine hydrochloride (Gu HCl)[42]. The detection was scored using real time monitoring with double-strand DNA binding dye. Each condition was tested with 24 replicates of ~10 copies of SARS-CoV-2 RNA. Comparing the number of positives across the different conditions, 75 μM PAPS dye and 100 μM $Mn^{2+}$, added individually or in combination, had no significant impact on the detection sensitivity. Moreover, not only was Gu HCl compatible with the reaction conditions, but also slightly increased the detection sensitivity in the case with E1 primer or E1 primer as described previously[42]. In the case with N2 + E1 dual primer set, addition of Mn and PAPS both resulted in increased detection frequency compared to the control condition, potentially indicating improved efficiency, but at least indicating no deleterious effect from their inclusion in the RT-LAMP reactions.

To demonstrate the advantages of the PAPS colorimetric reporter system over pH-based detection, we tested both and

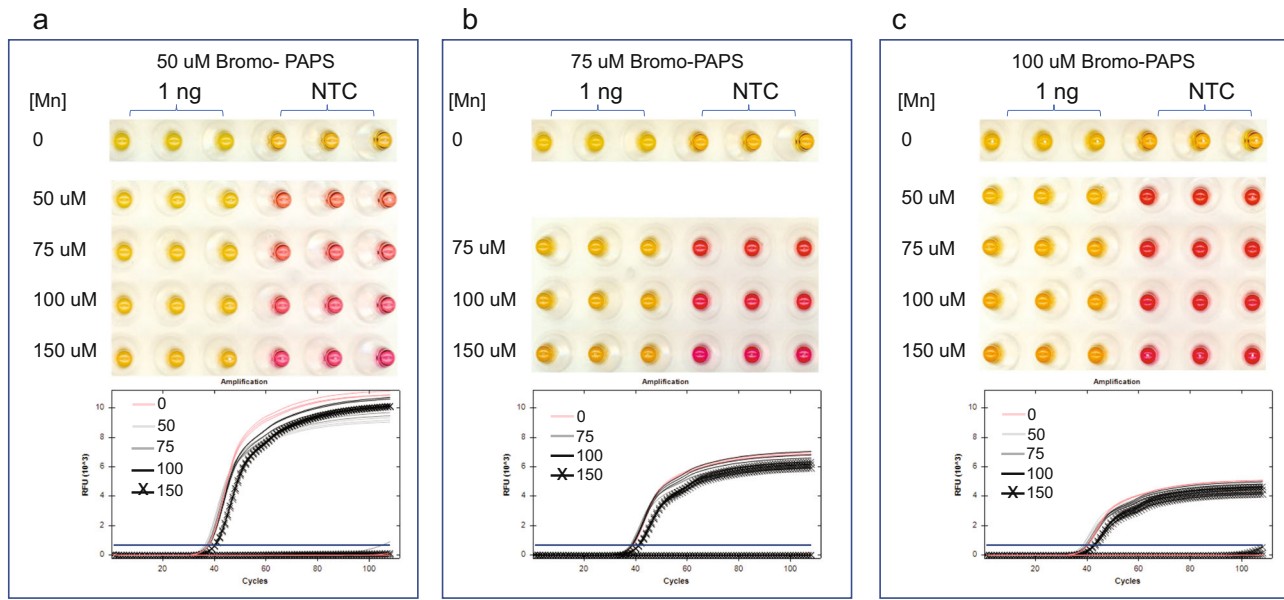

**Fig. 3 Determining optimal PAPS dye and $Mn^{2+}$ concentrations in LAMP.** Three concentrations of Bromo-PAPS were tested with 0, 50, 75, or 150 μM of $Mn^{2+}$. The colors of post-LAMP reactions are shown on the top, and the real-time curves on the bottom. The color coding for the real-time curves is the same as that of the labels for $Mn^{2+}$ concentrations. **a** 50 μM 5-Bromo-PAPS. **b** 75 μM 5-Bromo-PAPS. **c** 100 μM 5-Bromo-PAPS.

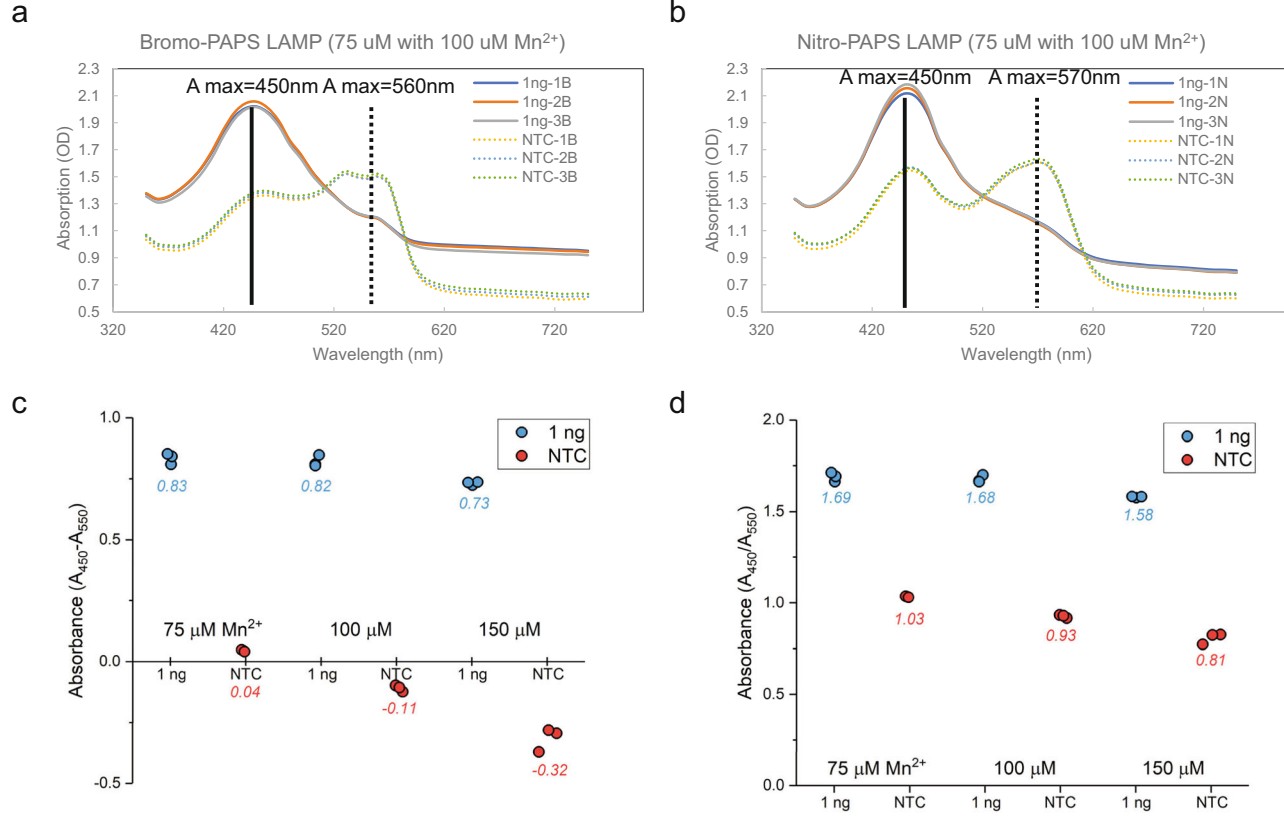

**Fig. 4 PAPS dyes absorbance spectrum shift during LAMP. a** Absorption measured from 360 to 750 nm wavelength in reactions with 75 μM 5-Bromo-PAPS and 100 μM $Mn^{2+}$. Reactions in triplicate with 1 ng of lambda DNA (1ng-1B, 1ng-2B, and 1ng-3B) or NTC controls (NTC-1B, NTC-2B, and NTC-3B). **b** 75 μM 5-Nitro-PAPS and 100 μM $Mn^{2+}$. Reactions in triplicate with 1 ng of lambda DNA (1ng-1N, 1ng-2N, and 1ng-3N) or NTC controls (NTC-1N, NTC-2N, and NTC-3N). **c** Relative gain of absorption at 450 and 550 nm in reactions with 75 μM 5-Bromo-PAPS and 75–150 μM $Mn^{2+}$. Positive reactions are graphed as blue circles and NTC as red with average ($N = 3$) as test label. **d** Ratio of absorption at 450 nm over 550 nm from reactions with 75 μM 5-Bromo-PAPS and 75–150 μM $Mn^{2+}$.

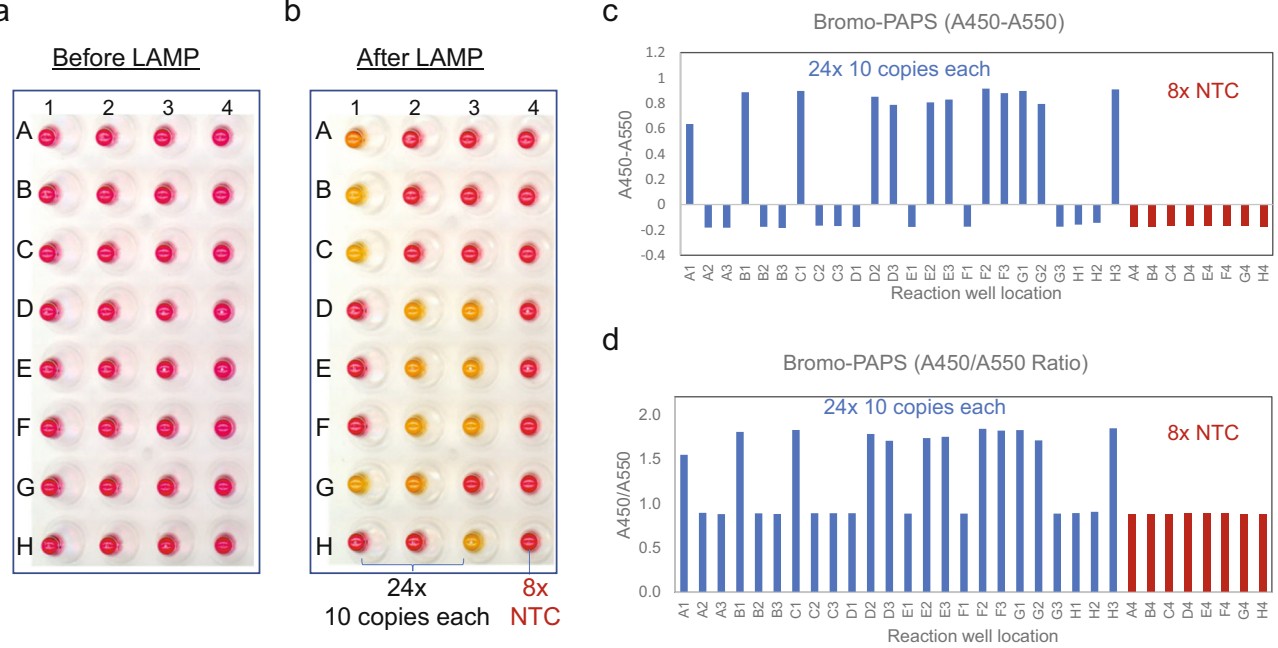

**Fig. 5 Detection of SARS-CoV-2 RNA by RT-LAMP using a PAPS dye as a reporter.** RT-LAMP reactions with E1 primer set are shown for 24 reactions, each with approximately 10 copies of target RNA (A1-H3) and 8 NTC reactions (A4-H4) in the presence of 75 μM 5-Bromo-PAPS and 100 μM $Mn^{2+}$. **a** Before amplification. **b** After amplification. **c** Relative gain of absorption at 450 and 550 nm. **d** Ratio of absorption at 450 over 550 nm.

**Table 1 Sensitivity of RT-LAMP in the presence of Mn$^{2+}$ and/or Bromo-PAPS.**

| Condition | Primer and Gu HCl | | Mn$^{2+}$ | | Mn$^{2+}$ |
|---|---|---|---|---|---|
| | | | | 5-Bromo-PAPS | 5-Bromo-PAPS |
| 1 | N2 | 6/24 | 5/24 (0.731) | 4/24 (0.477) | 7/24 (0.745) |
| 2 | N2 (+Gu HCl) | 10/24 | 8/24 (0.551) | 9/24 (0.768) | 9/24 (0.768) |
| 3 | E1 | 7/24 | 6/24 (0.745) | 7/24 (1.00) | 5/24 (0.505) |
| 4 | E1 (+Gu HCl) | 12/24 | 12/24 (1.00) | 13/24 (0.773) | 12/24 (1.00) |
| 5 | N2 + E1 (+Gu HCl) | 14/24 | 21/24 (0.023) | 19/24 (0.119) | 21/24 (0.023) |

The effect of 100 µM Mn$^{2+}$ and 75 µM 5-Bromo-PAPS on the sensitivity of RT-LAMP was tested using primer sets N2 and E1 alone or together without or with Gu HCl in a total of 5 conditions. Each test had 24 reactions with approximate 10 copies of synthetic SARS-CoV2 RNA and fractions of positives are shown. All positive reactions were determined by real time monitoring using dsDNA binding dye. Two proportion Z-test (two-tailed) was performed between test conditions (Mn$^{2+}$ and/or PAPS dye) and basal reactions and the probability that they are the same from the basal condition are shown in parenthesis. Reactions containing both 5-Bromo-PAPS and Mn$^{2+}$ were also scored based on color change and the results were identical to that by real-time monitoring.

compared their tolerance of various buffer carryover (Supplementary Fig. 3). First, we tested the effect of 0–7 µL (0–28% v/v) of a typical nucleic acid purification column elution buffer (Qiagen EB, 10 mM Tris pH 8.5) in 25 µL LAMP reactions. No interference in color change or inhibition of the LAMP reaction with the PAPS reporter system was observed under these conditions. However, with pH-based colorimetric detection, positive and negative reactions displayed much less contrast even with 3–4 µL of the elution buffer. Similarly, 0–7 µL of viral transport medium (VTM) was tolerated by PAPS-based colorimetric detection, whereas clear visual contrast was only achieved up to 4 µL of VTM with the pH-based system. These results demonstrate that PAPS-based detection permits the use of higher volumes of buffering agents, and thus overcomes a critical limitation associated with pH-based colorimetric LAMP[35]. With the PAPS-based mechanism, reactions could be affected by any sample containing significant concentrations of manganese ions or EDTA, so these factors should be considered in developing assays with field and direct sample applications. As an example, we investigated the effect by EDTA and found that, in the presence of 100 µM 5-Bromo-PAPS, only high concentrations of EDTA ( > 75 µM) started to affect the color change (Supplementary Fig. 4). However, increasing the Mn$^{2+}$ concentration could rescue the color change without sacrificing the LAMP reaction speed.

After realizing successful colorimetric detection of LAMP, we next tested PAPS dyes as visual reporters in PCR. Given the thermal cycling conditions and significantly reduced DNA yield relative to LAMP, previous attempts at colorimetric PCR have been unsuccessful (or at best very restricted) using pH-based dyes[18]. We tested various concentrations of 5-Bromo-PAPS and Mn$^{2+}$ and found that each at 50 µM produced the best visual contrast in the PCR mixes. This is slightly different from the best conditions for LAMP, likely due to the significant differences in Mg$^{2+}$, dNTP, and DNA yield in the different reactions. We amplified 0.5–5.0 kb long fragments from lambda DNA in the presence of 5-Bromo-PAPS and Mn$^{2+}$ with three commercial PCR master mixes (Fig. 6a). Before PCR cycling, all reactions displayed a red color as it was seen in the LAMP mix. After the completion of PCR (36 cycles), reactions containing a template DNA changed color from red to yellow, while NTC reactions remained red. Samples with color change matched exactly with those of successful standard PCR amplification, as determined by agarose electrophoresis (Fig. 6b). We did observe some minor initial color and color change variation among the three PCR mixes, likely reflecting differences in their composition, but also suggesting this system is generally compatible with commercial PCR mixes. The use of pH-dependent colorimetric dyes in PCR required overlay of mineral oil atop 50 µL PCR volumes and long (>1.0 kb) amplicons[18], but none of those restrictions is necessary for PAPS-based detection. In fact, standard PCR conditions and

amplicons as small as 0.5 kb all resulted in obvious color change using PAPS dyes as reporters.

To estimate the robustness and sensitivity of PAPS-based colorimetric PCR detection, we amplified DNA fragments from E. coli and human genomes for 32–44 cycles and compared the color change with the yield of specific product bands by electrophoresis (Fig. 6c). For a 1.0 kb E. coli fragment using 1 ng of input genomic DNA template (~200,000 copies), the reaction color changed to yellow in less than 32 PCR cycles. For a human fragment of the same size using 10 ng of genomic DNA (~2900 copies), it required about 40 cycles to change color. When the human fragment size increased to 2 kb, the cycle numbers required for the color change reduced to 32–36 cycles. These results indicated that the color change depended on the amount of DNA produced, corroborating the product yields visualized by gel electrophoresis. We further quantified the relationship between visual color change and DNA yield. We performed colorimetric PCR reactions with a range of cycle numbers amplifying low (human genomic DNA, ~2900 copies) and high (lambda DNA, ~$1.91 \times 10^{7}$ copies) copy number templates, including for each 0.5, 1.0, and 2.0 kb amplicons. The reactions were checked for visual color change, measured for both absorbance and DNA yield. The absorbance change (A450-A550) indeed correlated with DNA yield, and could be detected at least 4 cycles before the visible color change (Supplementary Figs. 5 and 6). As expected, a high template copy number for the lambda amplicons required fewer PCR cycles to accumulate similar amount of DNA and change color than that of low copy number human templates. The DNA yield at a point when color change starting to be perceivable by the naked eye were very similar across all amplicons of both lambda and human (Fig. 6d and Supplementary Fig. 7) in the range of 40–65 ng/µl.

As an example of a workflow improvement enabled by colorimetric PCR, we next sought to replace electrophoresis in a colony PCR experiment. Colony PCR is routinely used to identify colonies carrying a correct plasmid DNA assembly, directly from a small amount of E. coli cells. It typically requires manual gel electrophoresis to confirm the product presence and size. We chose cloned plasmids transformed in E. coli and analyzed the PCR performance using PAPS. First, as is commonly done for a novel PCR assay, we determined the best annealing temperature for each primer pair using purified plasmid DNA in the presence of 5-Bromo-PAPS and Mn$^{2+}$ (Fig. 7a). Then we compared the results to those using standard agarose gel electrophoresis. We found that the optimal conditions and PCR yields were much the same between the two methods. This indicates PAPS can also be applied to streamline the PCR optimization and avoid the need for a gel. We then performed PCR with cells from eight E. coli colonies, four each from two different plasmids. The selected primer pairs were either unique for the first 4 colonies (Fig. 7b), unique for the second 4 colonies (Fig. 7c), or common for both

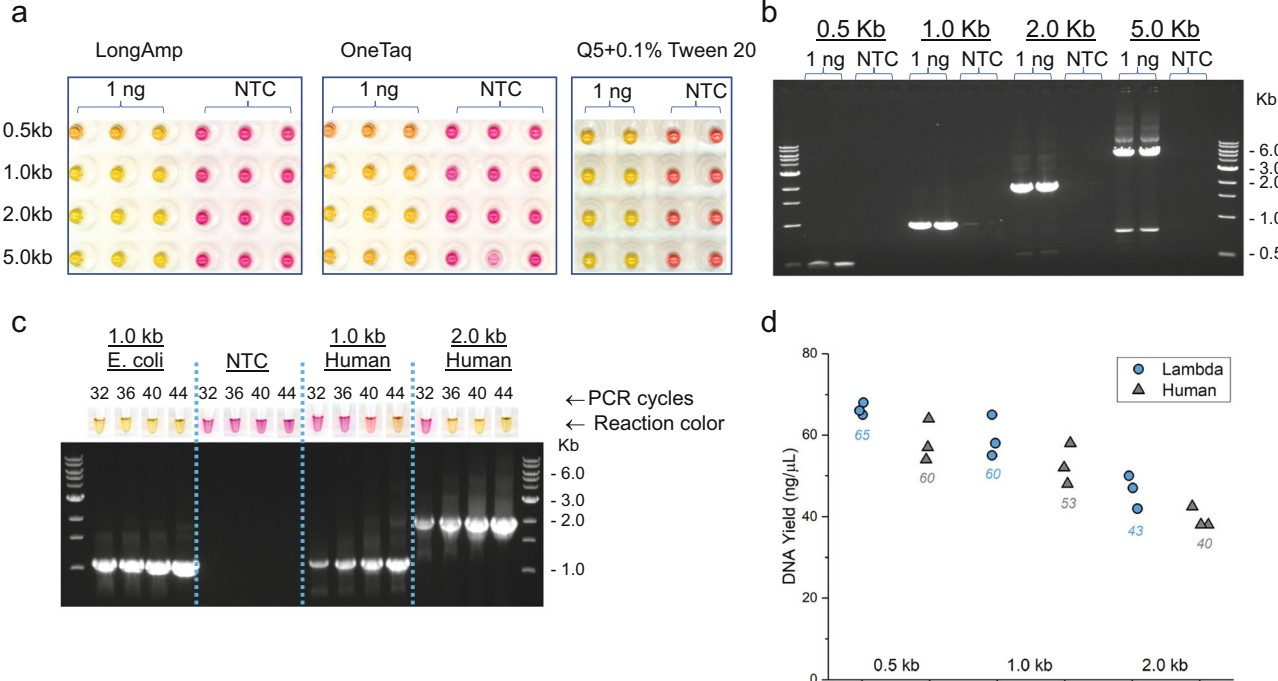

**Fig. 6 PAPS colorimetric detection of PCR amplification.** The reaction color after completion of PCR cycling in triplicates (LongAmp and OneTaq) or duplicate (Q5) is shown for amplifying 0.5–5.0 kb lambda DNA fragments or the corresponding NTC, in the presence of 50 µM Bromo-PAPS and 50 µM $Mn^{2+}$. **a** Reaction colors among reactions with and without DNA template after PCR with LongAmp *Taq*, OneTaq, and Q5 master mixes. **b** Agarose gel electrophoresis of PCR products generated by Q5 2× master mix. **c** Number of PCR cycles required for color change when amplifying DNA fragments from *E. coli* and human genomes. **d** DNA yield at the point of visual color change (A450 − A550 = 0.1) for lambda (blue circle) and human (gray triangle) amplicons with average ($N = 3$) value label.

(Fig. 7d). The resulting PCR reactions changed color only when there was a correct target plasmid and matched perfectly with the specific bands detected by agarose electrophoresis.

## Conclusions

We have utilized pyridylazophenol dyes for simple visual detection of amplification reactions. While multiple methods for this goal have been described, all of them, including our own previous work, have significant limitations in color contrast, buffer and media tolerance, or compatibility with PCR and other amplification methods. As shown here, the use of PAPS dyes and manganese ion overcomes these barriers and provides a strong visual readout of amplification in both isothermal and PCR applications, providing a more robust approach to universal colorimetric assay readout.

## Methods

All chemicals were purchased from Sigma-Aldrich, unless otherwise noted: 5-Bromo-PAPS (Sigma-Aldrich #93832), 5-Nitro-PAPS (Dojindo Molecular Technologies, #N031), 4-(2-pyridylazo)resorcinol (#178268), hydroxynaphthol blue (Acros Organics, #204880050), Eriochrome® Black T (#858390), MnCl₂ (#221279), CaCl₂ (#223506), CuCl₂ (#C3279), ZnCl₂ (#746355), FeSO₄ (#215422), NiCl₂ (#N6136), ReCl₃ (#309184), CrCl₂ (#450782).

All molecular reagents were from New England Biolabs (NEB), unless otherwise noted. For screening metal sensing dyes and metal ions, 25 µL LAMP reactions were performed in triplicate using WarmStart® LAMP Kit (DNA & RNA) (#E1700) or LAMP Kit with UDG (#E1708) with a lambda DNA primer set[18] with 1 ng lambda DNA (#N3011) or water no-template controls. LAMP reactions contained 20 mM Tris at pH 8.8 and standard concentrations of 1.4 mM each dNTPs and 8 mM MgSO₄. Double-strand DNA binding dye (#B1700S) was included for confirmation of amplification with real-time fluorescence. Dyes and metal ions were diluted to 25x concentration and then mixed into LAMP reactions to 1x concentration before incubation. The reactions including all components were incubated at 65 °C on a Bio-Rad CFX96 real-time PCR instrument with the fluorescence acquired at 15 second intervals (108 cycles, ~40 min). Before and after incubation, the color of each reaction was recorded by scanning with an office

scanner (Epson Perfection Photo Scanner V600). The absorbance of each reaction was measured on a SpectraMax M5. Thermostable Inorganic Pyrophosphatase (#M2926) was added (1 Unit) to 25 µL LAMP reactions in the test of pyrophosphate requirement. For comparing tolerance of carryover solutions, the elution buffer (10 mM Tris, pH 8.5) from a QIAquick PCR Purification Kit (Qiagen, #28104) and a viral transport medium (Universal Viral Transport BD, #220531) were used. The pH-based colorimetric LAMP reactions were performed with WarmStart® Colorimetric LAMP 2X Master Mix (DNA & RNA) (M1800) following the product recommendations. For RT-LAMP, the SARS-CoV-2 E1 and N2 primer sets from our previous study[42] were used along with synthetic SARS-CoV-2 RNA from Twist Bioscience (Twist Synthetic SARS-CoV-2 RNA Control 2, #102024) diluted to ~10 copies/µL in the presence of 10 ng/µL Jurkat total RNA.

Colorimetric detection of PCR amplification was performed in 25 µL reactions by including 50 µM PAPS dye and 50 µM $Mn^{2+}$ into reactions with either LongAmp® Taq 2× Master Mix (#M0287) or OneTaq® 2X Master Mix with Standard Buffer (#M0482). The same concentrations of PAPS dye and $Mn^{2+}$ were added to Q5® High-Fidelity 2× Master Mix (#M0492) and supplemented with Tween 20 to a final concentration 0.1%. The primer pairs for amplifying lambda DNA fragments had the same forward primer (5′-CCTGCTCTGCCGCTTCACGC) with different reverse primers to create different product lengths (0.5 kb, 5′-TCCGGATAAAAACGTCGATGACATTTGC; 1.0 kb, 5′-GATGACGGCATCCTCACGATAATATCCGG; 2.0 kb, 5′-CCATGATTCAGTGTGCCCGTCTGG and 5.0 kb; 5′-CGAACGTCGCGCAGAGAAACAGG). PCR conditions for producing these Lambda fragments using either LongAmp or OneTaq were: 94 °C 30″; 35 cycles of 94 °C 15″, 55 °C 30″ and 65 °C 2′; 65 °C 10′. Conditions for Q5 were: 98 °C 30″; 35 cycles of 98 °C 10″, 70 °C 30″ and 72 °C 2′; 72 °C 5′. For amplifying 1–2 kb fragments from *E. coli* and 0.5–2 kb human genomic DNA, reactions were performing using LongAmp® Hot Start Taq 2X Master Mix (M0533) with 50 µM Bromo-PAPS, either 25 or 50 µM $Mn^{2+}$, and either 1 ng *E. coli* DNA (~200,000 copies) or 10 ng Jurkat cell DNA (~2,900 copies). Cycling conditions were: 94 °C 30″; 16–44 cycles of 94 °C 15″, 57 °C 30″, and 65 °C 100″. A final 65 °C 10′ step was only used for the 44× cycle condition. The primer sequences were: 1 kb *E. coli* fragment (Forward: 5′-CCTGGATCCAGATGCAGTAATACCGC, Reverse: 5′-TCCGAGGATGGTATTCGTCATG); 0.5 kb human fragment (Forward: 5′-GGGGCACCTTCTCCAACTCATACT, Reverse: 5′-CGAGCTACCACGCAGACATCAACC); 1 kb human fragment (Forward: 5′ GGGGCACCTTCTCCAACTCATACT, Reverse: 5′-CCTCATTTGGGGAGGGGTTATCT); 2 kb human fragment (Forward: 5′-GAAGAGCCAAGGACAGGTAC, Reverse: 5′-CCTCCAAATCAAGCCTCTAC). DNA yield for PCR reaction was determined using Quant-iT™ PicoGreen™ dsDNA Assay Kits (Thermo-Fisher, P7589).

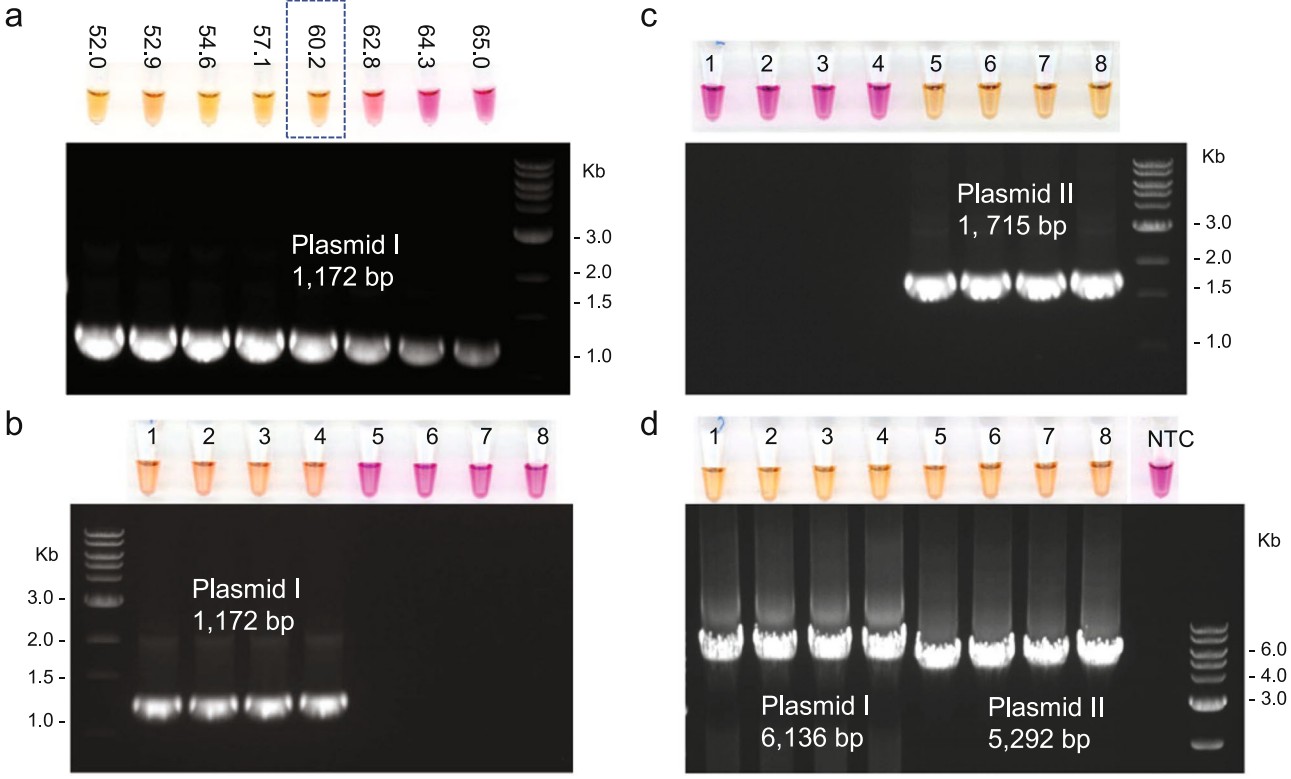

**Fig. 7 Application of colorimetric PCR for identification of *E. coli* colonies.** PCR reactions shown with color on the top and the corresponding lanes in the agarose gel image. Four colonies for plasmid I (1–4) and four for plasmid II (5–8) are shown to amplify with primers specific for one or both plasmids. **a** Optimization of annealing temperature (°C) for amplifying an 1172 bp fragment from plasmid I with purified plasmid DNA. The annealing temperature is shown above reaction tubes. **b** PCR with bacterial cells using primers recognizing plasmid I. **c** PCR with bacterial cells using primers recognizing plasmid II. **d** PCR with bacterial cells using primers recognizing both plasmids but generating different fragment sizes.

For colony PCR, a small portion of an *E. coli* colony was transferred using a pipette tip to 15 μL water. In all, 1 μL of the bacterial suspension was used for each PCR reaction with LongAmp. The cycling conditions for the 1.2 and 1.7 kb fragments were: 94 °C 30″; 35 cycles of 94 °C 15″, 60 °C 30″, and 65 °C 90″; 65 °C 10′. Similar conditions were used for amplifying the 6.1 kb and 5.3 kb fragments, except that the extension time at 65 °C was adjusted to 5 min. Also, 5 μL of each PCR product was analyzed by electrophoresis on 1% agarose gel along with a Quick-Load® 1 kb DNA Ladder (#N0468).

**Statistics and reproducibility.** All data were collected with at least $N = 3$ replicates, and where appropriate average values are reported above. For assessing LAMP performance below the LOD of the SARS-CoV-2 assay, $N = 24$ was used and all reactions shown.

**Reporting summary.** Further information on research design is available in the Nature Research Reporting Summary linked to this article.

## Data availability

All data generated or analyzed during this study are included in the published article and its Supplementary Data.

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

## Author contributions

Y.Z., E.A.H., E.T., I.R.C., and N.A.T. conceptualized the study and edited the manuscript. Y.Z., E.A.H., and E.T. conducted experiments and collected data. Y.Z., I.R.C., and N.A.T. wrote the manuscript.

## Competing interests

The authors declare the following competing interests: all are employees of New England Biolabs, manufacturer of enzymes and reagents described in the text. Authors are named inventors on patent applications related to the work described herein.
