## [Peer Review File · Communications Biology]

Reviewers' comments:

Reviewer #1 (Remarks to the Author):

The authors investigated the use of PAPS dyes for visual colorimetric detection of amplified DNA. This work can be impactful to support the implementation of field-friendly diagnostic tools in point-of-care and home settings. The developed system appears to be useful in detecting DNA that has been amplified using both PCR and isothermal amplification techniques, though, additional results are needed to show both compatibility with a variety of amplifications kits/assays commonly employed and to describe how the system would be practically implemented.

General comments:

-Figures need to be reworked, changes in both the visual format and plot choice plus their finish are needed.

-There should be a more significant description of the system's specificity. Little is done to show that the system isn't prone to false negatives/positives from other compounds that might compete or interfere.

-There should be a description for how this system could actually be implemented for the intended use. The concentrations of Mn^{2+} and the dye used were found to be important for producing the expected result, so how would these be managed if used in a field-friendly detection system?

-A key limitation is not describing the colorimetric change in a meaningful way outside of classical analytical instrumentation. Work is needed to show that the visual color change could be measured using simple, accessible tools like image or color analysis, and show that these approaches are implantable for routine analysis

In the SI: The layout of Figure S1 should be similar to that of what is shown in Figure 1 in the main text. At present it is difficult to flip between A&B to see the colorimetric change. The middle column labels should also be moved to the left side to make them easier to see.

In the SI: The arbitrary threshold can and should be based on some kind of field-friendly tool (color/image analysis) that can be implemented and controlled for.

Line items:

Line 35: What is this instrumentation? Needs to be given to appropriately understand the increase in cost/complexity

Line 163: Would it be possible for any other compounds, especially those commonly found in PCR/isothermal assay kits, to cause a similar effect? EDTA is a relatively common nucleic acid buffer and it is important to qualify how often you might see this effect with EDTA or other compounds

Line 171:

How was optimal visual contrast defined? Objective or subjective? Visual colors are highly dependent on the individual viewer, what one sees as different others may not. More should be done to qualify what the optimal visual response is, and this should be reported by some kind of field-friendly tool (e.g. image or color analysis)

Line 179: What are "objective automated measurement systems"? Need to be explained

Line 192: The 50% accuracy is not great for this test given it is at 50% in the ideal conditions (controlled lab environment with known inputs and access to the supplies/tools needed). A lot more needs to be done to qualify why this result is so low, and if it is a constraint of the developed detection system or a limitation from the amplification system used

Table 1 needs to be significantly improved with regards to listing out what each acronym means, as it currently stands you have to move back in forth in the paper to determine the acronyms and what they are with some acronyms (N2 and E1) never being properly explained

In Table 1, the note after the asterisk makes the understanding of the tables results actually more difficult that if it weren't included. It would be best to just say that the results were all determined by real time monitoring. Outside of the figure in the written text there should be a clear description that all results were measured with real time monitoring and that the final test (both with the colorimetric probe and Mn) was also scored based on the visual colorimetric response

Line 201: These results clearly illustrate that the dye itself is not problematic (i.e. the reactions without the dye have similar % positives as the ones with the dye) BUT it desperately needs to qualify why the results are low across the board. Under the best case scenario (all reagents, in a lab setting with controlled tools and supplies) there is only 88% sensitivity, which is an issue

Line 204: In one scenario the number of positives increase by $n=7$, a 40% increase which is not slight. A better explanation of why this increase would be observed is needed, given the developed colorimetric probe should work independent of the pH-based probe

Line 206: The experimental design of this section is clearly biased to produce poor results for pH probes in comparison to this dye that is known to not have pH sensitivity. While I think this information would be important to include in the SI, this space would be much better used to better qualify the developed system and determine the specificity of the system against other reagents/compounds commonly found in nucleic acid amplification techniques

Line 211: This needs to be better explained & qualified. Again, visual contrast based on an individuals perception of color is alone not enough to define a detection method if it were applied to a setting where access to real time monitoring or instrumental absorbance measurements are limited

Line 215: If this has been tested, the results should be included. Otherwise, this statement should be left out as it is not qualified as to how it can be assumed

Line 233: Why weren't pH probes compared again here? The start of the section explains that they are more difficult to use in PCR compared to LAMP, but jt still needs to be shown how they actually compare to illustrate that the developed system is an improvement

Line 255: This concentration is high compared to the instrumental methods that are used for nucleic acid quantification. Standard absorbance measurements for example have detection limits in the low (less than 10) ng/uL and certain platforms (i.e. Nanodrop) are able to achieve sub ng/uL concentrations. While this approach cannot reasonably be compared to instrumental methods given that it is intended to be used in non-lab settings, it is also important to describe how these concentrations compare to existing techniques and methods (both those that are field-friendly and lab-based)

Reviewer #2 (Remarks to the Author):

Major Claims of the Paper-

This manuscript details a novel colorimetric amplification detection for LAMP utilizing pyridylazophenol dyes binding manganese ion, Mn^{2+} . There is a need for a robust, low sophisticated instrumentation, pH independent, and strong visible color change in response to nucleic acid amplification within this loop-mediated isothermal amplification (LAMP) method to make this method more readily available. The author screened metal-sensing dyes for their ability to meet these goals. They found two related dyes, PAPS, that in the presence of Mn^{2+} produced a color change difference between a positive reaction and no template control reactions. This colorimetric detection worked robustly for both LAMP and PCR amplification. In addition, by taking the absorbance difference (A450-A550) and/or the ratio (A450/A550), a quantitative read-out accurately correlates with DNA yield.

The claims of this paper are novel and will be of interest to others in the application of LAMP and PCR methods. The colorimetric results are novel and will be of interest in the wider field of chemical biology.

Questions and concerns:

Does your assay require the opening of tubes to add the Mn^{2+} ? Is there a concern regarding contamination in your technique? According to the methods (line 88 and 89), "dyes and metal ions were diluted to 25x concentration and then mixed into LAMP reactions to 1x concentration before incubation", what is the time between red to yellow color change in positive reactions? What is the pH of your reactions before LAMP and after LAMP? I realize you are using a strong buffer but did you take the pH to be sure the reaction and/or addition of PAPS and metals did not exceed your buffer capacity? For example, the color of HNB changes depending on the pH of the solution. Does the PAPS color change depending on the color of solution? Ref 36 indicates that the PAPS chelate metal ions at optimum pH ranges. It would be interesting to see the pH dependency of Mn^{2+} with the PAPS. Ref 37 show 5-Br-PAPS chelating Co^{2+} at a particular pH (pH 7.0) with the addition of PDADMAC. This paper also suggests the structure of 5-Br-PAPS at pH 7 in the deprotonated phenol. In addition, Co^{2+} and 5-Br-PAPS in absence of PDADMAC caused a color change from yellow to red purple.

Any concerns about non-template amplification?

Does the addition of pyridylazophenol dyes and manganese ion effect the LAMP reaction? Noticed the PAPS dyes and Mn^{2+} was added to the reactions prior to amplification which should prevent any contamination in the methods. But in the results, stated line 143 and 144 "when added to the amplification reaction mixture" is this after the amplification? If so are the tubes opened for the addition of Mn^{2+} after the amplification process while the 5-Br-PAPS is in the buffer solution before the amplification process?

Are the Pyridylazophenol dyes pH sensitive? Was the pH determined before and after the reaction to ensure there were no pH effects with this dye?

Line 6 a comma is needed "but as the amplification techniques have moved away from the lab, complementary detection"

Line 11 manganese ion

Line 40 a comma and not a dash "large DNA yield-or the byproducts of its synthesis, can be exploited"

Line 53 no dash needed for "in a weakly-buffered solution"

Line 66 and 67 the N should be italicized in the name "2-(5-Bromo-2-pyridylazo)-5-[N-propyl-N-(3-sulfopropyl)amino]phenol" and "2-(5-Nitro-2-pyridylazo)-5-[N-n-propyl-N-(3-sulfopropyl)amino]phenol"

Line 96 within the Method: "elution buffer (10 mM Tris, pH8.5)" needs a space between pH 8.5

Line 143 was Mn²⁺ added to the reaction mixture after the amplification process? Were the tubes opened in order to add the Mn²⁺? In the methods section, line 89 "concentration before incubation."

Line 146 add the word color "red color (Figure 1A, Supplementary Figure 1)." How much time did it take for the color change to occur?

How long does it take for the color to change from red to yellow in the positive reactions? For example reference 17 using calcein and Mn²⁺ the visualization takes 30-60 min.

Line 147 for the 5-Nitro-PAPS color change "back to yellow upon successful LAMP amplification" how much time did it take for this color change to occur?

Line 155 add a comma and a transition such as "or other metal ions, and, thus accordingly focused"

Line 158 need to add 's to dye "thereby restoring the dye's original yellow color (Figure 2)."

Line 160 add (PPase) for "adding inorganic pyrophosphatase (PPase), which converts"

Line 163 spell out the chemical name for ethylenediaminetetraacetic acid (EDTA)

Line 285 Figure 1 "lambda DNA (1ng)" needs a space between 1 ng. Why is the color different for the 100 μM and the 800 μM concentrations of Mn²⁺? Notice that there is no color change at the higher concentrations of Mn²⁺ (800 μM). The Real time graphs of RFU verses cycles are slightly blurry. Looks like the amplifications using the higher concentrations 800 μM Mn²⁺ didn't produce any RFU units until greater than 80 cycles. RFU y-axis is labeled in Figure 1E differently than the others.

Line 294 Suggestion for Figure 2A to structurally follow the schematic 2C where the PAPS chelate the Mn²⁺; would the phenol be deprotonated and negatively charged like that in papers references 37 and 38? Did you take the pH of the samples before LAMP and after in the presence of Mn²⁺ to confirm there was no pH changes? Maybe interesting to see how the color of the PAPS change when chelating Mn²⁺ in a pH study similar to the referenced papers 37 and 38.

Line 300 Figure 3B why is there no data shown for 50 μM of Mn²⁺?

Line 167 In the paper state "Concentrations of each dye from 50-100 uM" however only Br-PAPS is shown in Figure 3.

Line 307 Figure 4. A, Absorption measured from 360 to 750 nm wavelength however your x-axis shows the measurement to 720 nm

Suggestion: Change Mn⁺⁺ in figures with Mn²⁺ to match manuscript

Line 307 Figure 4. D, also with concentrations with 75 uM 5-Bromo-PAPS and 75-150 uM Mn²⁺ should be added to legend. What is the 1ng-1B, 1ng-2B, and 1ng-3B in Figure 4? Assuming reactions in triplet?

Table 1 suggestion change Mn⁺⁺ to Mn²⁺ and define N2, change +GuCl to GuHCl to be consistent, define E1 and E1700.

Line 209 add a space "Quiagen EB, 10mM Tris pH 8.5)" to 10 mM

Line 215 "Tolerance in quantitative measurement systems, e.g., absorbance, is likely" remove the commas and add parentheses, measurement systems (e.g. absorbance) is likely

Line 226 and 227 no dash for the no template control reactions or could use NTC

Line 231 What are the standard PCR mixes (e.g. and list them)? Are these the three chosen in Figure 6A?

Line 276 and 277 perhaps you should also include manganese ion "the use of PAPS dyes and manganese ion" overcome

Line 338 Figure 7. "the annealing temperature is shown" should include °C perhaps in parenthesis after temperature (°C)

Line 339 B "bacterial cells using primes recognizing plasmid I." should be primers?

Line 339 C "bacterial cells using primes recognizing plasmid II." Should be primers?

Line 340 D "bacterial cells using primes recognizing both plasmids" Should be primers?
Suppl. Figure 1B and 1C

Line 2 the chemical formulas should include subscripts. Did you mean to use iron(II) sulfate and not chloride?

Suppl. Figure 2:

Line 12 In the legend "visual detection is shown for 5-Bromo-PAPS (PAPS)" add (PAPS) to be consistent with images. Also replace Mn⁺⁺ with Mn²⁺

Suppl Figure 3:

Line 22 Space between "1ng lambda DNA" to 1 ng

Line 23 Space between "0.5kb amplicon" to 0.5 kb

Line 24 and 25 Space between "450nm and 550nm" to 450 nm and 550 nm

In A for both Suppl Figure 3 and Suppl Figure 4 should also have spaces between 0.5 kb, 1.0 kb, 2.0 kb

References

Reference 5 line 354, reference 7 line 359, reference 12 line 372, reference 23 line 402, reference 25 line 407, reference 33 line 430 remove et al and include all the authors

Reference 37 line 429 and reference 39 line 445 superscript Cu²⁺

Reviewer #3 (Remarks to the Author):

COMMSBIO-22-1745-T

Recommendation: Accept with major changes.

This manuscript describes an indicator for detection of amplification of DNA by PCR and RT-LAMP

methods. The test detects the hydrolysis of the nucleotide triphosphates concomitant with DNA polymerization. The idea is to carry out the amplification in the presence of 50 micromolar Mn(II) and 50 micromolar Mn(II) indicator dye. Manganese(II) binds strongly to the inorganic pyrophosphate (PPi) formed during DNA polymerization, so when the concentration of the ds amplicon rises into the micromolar range, enough PPi is formed to pull the Mn(II) off the dye, and the dye changes color. The work is solid, the materials and methods are described adequately, and its immediate applicability to NAT visualization makes it suitable for the readership of Comm. Bio. I'd be interested in giving it a go.

→This method provides a simple visual mechanism for detection of a NAT test result.

→The method is superior to the pH indicator dyes that are used in commercial RT-LAMP COVID tests because it tolerates carryover of buffers and can be used on matrices with a wide range of pH values.

The authors screened several Mn(II) indicator dyes and found one with a strong color change (red to yellow). The suggested mechanism for the color change was tested by adding an enzyme to hydrolyze PPi to phosphate (which does not bind Mn(II) as strongly) and by using EDTA to chelate the Mn(II). They tested the method for detection of RT-LAMP amplification of SARS-COV2 RNA, finding performance comparable to that of RT-LAMP with pH indicator dyes, and higher tolerance to carryover from sample prep stages. The mss claims that the dye and Mn(II) had no impact on the detection sensitivity, I think a little more work is needed to establish this (see change #1).

They then set up a variety of PCR amplifications (lambda DNA of several lengths, E. coli and human DNA fragments) using different polymerases and monitored the DNA production with a conventional fluorescent dsDNA dye. The utility of the dye test for monitoring plasmid transfection in place of gel electrophoresis is nicely shown in Figure 7.

Major changes:

1) The data in table 1 show positives and negatives detected by a conventional DNA fluorescent dye from 24 samples at analyte concentrations near the LOD for the RT-LAMP test; results range from 4/24 to 21/24 positives. It's hard to see which differences are significant and which are due to random error. It would strengthen this data to show a replication and some statistical analysis.

Minor changes:

2) The optimal concentrations of the dye and Mn(II) are different for LAMP and for PCR, please add some explanation in the text.

"For subsequent studies we chose 75 μ M PAPS dye with 100 μ M Mn²⁺ for optimal visual contrast." In materials and methods, "Colorimetric detection of PCR amplification was performed in 25 μ L reactions by including 50 μ M PAPS dye and 50 μ M Mn²⁺ into reactions"

3) typos

a) line 200 and 203 use GnHCl to abbreviate guanidinium HCl.

b) Figure 7 caption says primes instead of primers.

Regards,

Marya Lieberman
Department of Chemistry and Biochemistry
U. Notre Dame

We thank the reviewers for their thorough and critical examination of our manuscript. As noted below we have made extensive changes and additions as requested, with the revised version a much stronger and clearer presentation of the work.

Reviewers' comments:

Reviewer #1 (Remarks to the Author):

The authors investigated the use of PAPS dyes for visual colorimetric detection of amplified DNA. This work can be impactful to support the implementation of field-friendly diagnostic tools in point-of-care and home settings. The developed system appears to be useful in detecting DNA that has been amplified using both PCR and isothermal amplification techniques, though, additional results are needed to show both compatibility with a variety of amplifications kits/assays commonly employed and to describe how the system would be practically implemented.

General comments:

-Figures need to be reworked, changes in both the visual format and plot choice plus their finish are needed.

To address this point we updated all the figures, changed all the label fonts, and generally tried to make them all clearer and more legible.

-There should be a more significant description of the system's specificity. Little is done to show that the system isn't prone to false negatives/positives from other compounds that might compete or interfere.

-There should be a description for how this system could actually be implemented for the intended use. The concentrations of Mn²⁺ and the dye used were found to be important for producing the expected result, so how would these be managed if used in a field-friendly detection system?

In every experiment we include both positive and negative controls that were conducted in parallel and monitored with a dsDNA binding fluorescent dye for an orthogonal detection mechanism specifically to catch any false positives or negatives. As example sample matrices we used a nucleic acid column elution buffer and viral transport media, with significantly higher sample tolerance with the PAPS dye vs. the standard pH-based detection (now Supplementary Figure 3). Compounds that could interfere certainly include manganese and EDTA, and we show the effects of these in demonstrating the mechanism of detection. As noted below we also added Supplementary Figure 4 to specifically examine the tolerance to EDTA and its relationship with Mn²⁺ concentration (see below) We clarified these descriptions in the Discussion and added a sentence to describe limitations of sample tolerance for field use:

(Line 242) "With the PAPS-based mechanism, reactions could be affected by any sample containing greater than 75 μM manganese ions or EDTA (Supplementary Figure 4), so these factors should be considered in developing assays with field and direct sample applications. As an example, we investigated the effect by EDTA and found that, in the presence of 100 μM 5-Bromo-PAPS, only high concentrations of EDTA ($> 75 \mu\text{M}$) started to affect the color change (Supplementary Figure 4). However, increasing the

Mn²⁺ concentration could rescue the color change without sacrificing the LAMP reaction speed”

-A key limitation is not describing the colorimetric change in a meaningful way outside of classical analytical instrumentation. Work is needed to show that the visual color change could be measured using simple, accessible tools like image or color analysis, and show that these approaches are implantable for routine analysis

The overarching intent for the method is of course to provide a simple mechanism for detection, and we thought this was clear from the use of simple pictures of the reaction tubes for the majority of the data; use of absorbance measurement was an additional step for quantitative and objective analysis as has been required for EUA SARS-CoV-2 assays using the pH-based colorimetric LAMP. Use of simple image analysis tools is certainly possible, and we added text highlighting this with references at Line 190:

“For simple applications using smartphone cameras or basic image analysis, the processes and tools developed for colorimetric LAMP such as the LAMP Plate Reader app (<https://apps.apple.com/us/app/lamp-plate-reader/id1529271060>) and hue/saturation analysis (<https://pubmed.ncbi.nlm.nih.gov/33559876/>, <https://pubmed.ncbi.nlm.nih.gov/33299153/>) can be used for any of the colorimetric dyes and are suitable for PAPS.”

In the SI: The layout of Figure S1 should be similar to that of what is shown in Figure 1 in the main text. At present it is difficult to flip between A&B to see the colorimetric change. The middle column labels should also be moved to the left side to make them easier to see.

This figure has been adjusted as suggested and we thank the reviewer for noticing the inconsistency.

In the SI: The arbitrary threshold can and should be based on some kind of field-friendly tool (color/image analysis) that can be implemented and controlled for.

The threshold we use is indeed arbitrary, but was based on matching a visual indication of color change as multiple users could identify a change from the initial pink color. As described above use of quantitative analysis from absorbance or color analysis could certainly be used for calling positives, but this will vary with the tool and assay and we do not feel we should prescribe a standard to use here.

Line items:

Line 35: What is this instrumentation? Needs to be given to appropriately understand the increase in cost/complexity

We added the clarification “similar to real time qPCR machines” to this line of text.

Line 163: Would it be possible for any other compounds, especially those commonly found in

PCR/isothermal assay kits, to cause a similar effect? EDTA is a relatively common nucleic acid buffer and it is important to qualify how often you might see this effect with EDTA or other compounds.

EDTA can indeed be present in various buffers as a preservative, though generally at low concentrations. As we describe here tens of μM are needed to affect the PAPS signal and while it is a consideration to be aware of the concern is no different than general PCR or LAMP reactions as EDTA of course chelates Mg and can inhibit polymerases generally. To address this point we added Supplementary Figure 4 demonstrating EDTA effects and balancing with Mn^{2+} and describe it in the text in Line 245:

“As an example, we investigated the effect by EDTA and found that, in the presence of 100 μM 5-Bromo-PAPS, only high concentrations of EDTA ($> 75 \mu\text{M}$) started to affect the color change (Supplementary Figure 4). However, increasing the Mn^{2+} concentration could rescue the color change without sacrificing the LAMP reaction speed.”

Line 171: How was optimal visual contrast defined? Objective or subjective? Visual colors are highly dependent on the individual viewer, what one sees as different others may not. More should be done to qualify what the optimal visual response is, and this should be reported by some kind of field-friendly tool (e.g. image or color analysis)

The visual contrast is indeed very subjective and there could be differences in judgment across different individuals. However we are comparing two visual methods against each other, specifically for use in instrument-free simple diagnostics. “Optimal” visual response is also entirely subjective and we outline one way to quantitate that using absorbance in the next paragraph and data figures, with example thresholds and analysis metrics. Reporting with a tool or device would definitely be recommended for a diagnostic test or assay but that is beyond the scope of what we are describing here and should be left to the test or instrument developer.

Line 179: What are "objective automated measurement systems"? Need to be explained

We added “such as a spectrophotometer or plate reader” to this section.

Line 192: The 50% accuracy is not great for this test given it is at 50% in the ideal conditions (controlled lab environment with known inputs and access to the supplies/tools needed). A lot more needs to be done to qualify why this result is so low, and if it is a constraint of the developed detection system or a limitation from the amplification system used.

The test does not have a 50% accuracy. We are intentionally using an established RT-LAMP assay (see among many examples: <https://pubmed.ncbi.nlm.nih.gov/34476394/>, <https://pubmed.ncbi.nlm.nih.gov/33503071/>, <https://pubmed.ncbi.nlm.nih.gov/35617204/>) below its limit of detection to determine any difference in performance between the different detection schemes. We had stated this in the text but extended this description in Line 203:

“We performed 24 reactions with input of approximately 10 copies of viral RNA per reaction, which is below the limit of detection of a commercially available kit based on a pH-dependent dye (~ 50 copies, SARS-CoV-2 Rapid Colorimetric LAMP Assay Kit, NEB #E2019, ^{40,43})

and Line 215:

“We performed tests with N2 or E1 primer sets for SARS-CoV-2 using either a single set or both sets, with or without the pH-based colorimetric LAMP stimulator guanidine hydrochloride (Gu HCl) ⁴⁰. The detection was scored using real time monitoring with double-strand DNA binding dye. Each condition was tested with 24 replicates of ~10 copies of SARS-CoV-2 RNA. Comparing the number of positives across the different conditions, 75 μ M PAPS dye and 100 μ M Mn²⁺, added individually or in combination, had no significant impact on the detection sensitivity. Moreover, not only was Gu HCl compatible with the reaction conditions, but also slightly increased the detection sensitivity in the case with E1 primer or E1 primer as described previously ⁴⁰. In the case with N2+E1 dual primer set, addition of Mn and PAPS both resulted in increased detection frequency compared to the control condition, potentially indicating improved efficiency, but at least indicating no deleterious effect from their inclusion in the RT-LAMP reactions.”

Table 1 needs to be significantly improved with regards to listing out what each acronym means, as it currently stands you have to move back in forth in the paper to determine the acronyms and what they are with some acronyms (N2 and E1) never being properly explained

We updated Table 1 as suggested to clarify the comparisons and added more to the descriptive text to make sure the Table is more easily interpreted.

In Table 1, the note after the asterisk makes the understanding of the tables results actually more difficult that if it weren't included. It would be best to just say that the results were all determined by real time monitoring. Outside of the figure in the written text there should be a clear description that all results were measured with real time monitoring and that the final test (both with the colorimetric probe and Mn) was also scored based on the visual colorimetric response

We agree this could have been clearer and adjusted the table and text to match the suggestions.

Line 201: These results clearly illustrate that the dye itself is not problematic (i.e. the reactions without the dye have similar % positives as the ones with the dye) BUT it desperately needs to qualify why the results are low across the board. Under the best case scenario (all reagents, in a lab setting with controlled tools and supplies) there is only 88% sensitivity, which is an issue.

As described above we are not presenting a test here, and even so we're using 10 copies of viral RNA which is a very low input amount that even many PCR assays would not do much better with. The performance of this particular RT-LAMP assay has been described extensively, validated and used for large numbers of SARS-CoV-2 tests but that is not the message from this part of the manuscript, we are simply comparing the PAPS detection to the established pH-based approach at an input where differences in detection would be apparent. If we repeated this analysis with 100 copies then we'd show 100% sensitivity but that would be less useful to establish that PAPS+Mn is working as well as phenol red LAMP, as that's a much easier reaction to do. As noted above we added text to clarify this point.

Line 204: In one scenario the number of positives increase by n=7, a 40% increase which is not slight. A better explanation of why this increase would be observed is needed, given the developed colorimetric probe should work independent of the pH-based probe

We are not comfortable claiming this increase is caused by the addition of the PAPS dye, as the more fair comparison would be to 14 vs. 19 and 21 vs 21 out of 24, the equivalent conditions with and without Mn. The increase in sensitivity from the use of dual primer sets is consistent with our and others' previous reports, as is the increase from addition of guanidine hydrochloride. As suggested by Reviewer 3 we added a statistical analysis to this data (see below) and indeed in most conditions it appears the differences are indistinguishable from chance; we added text noting the one condition (dual primer+GnHCl) where the difference is significant, and while we do not claim the Mn/PAPS addition increases sensitivity it helps our argument that the detection is not adversely affected by their addition.

Line 206: The experimental design of this section is clearly biased to produce poor results for pH probes in comparison to this dye that is known to not have pH sensitivity. While I think this information would be important to include in the SI, this space would be much better used to better qualify the developed system and determine the specificity of the system against other reagents/compounds commonly found in nucleic acid amplification techniques.

We disagree that this data is "biased" it is simply demonstrating that carryover of common diagnostic sample media is a more significant problem for pH-based LAMP than the PAPS mechanism we are describing. Elution buffers and viral transport media are very common reagents found in nucleic acid amplification techniques so we think making this point as we do here is relevant to the method and its utility, as tolerance/effects of those things is an extremely common shortcoming of the pH-based method that we are overcoming.

Line 211: This needs to be better explained & qualified. Again, visual contrast based on an individual's perception of color is alone not enough to define a detection method if it were applied to a setting where access to real time monitoring or instrumental absorbance measurements are limited

If access to real time monitoring or absorbance measurement is limited, then the visual detection mechanism is the most likely and obvious way reactions would be judged. This has been the case for pH-based LAMP tests in field and resource-limited settings with scoring based on visual color an extremely common method. With this data we are simply comparing visually the PAPS vs. pH-based LAMP reaction with clearly better color contrast in the presence of higher amounts of sample using PAPS.

Line 215: If this has been tested, the results should be included. Otherwise, this statement should be left out as it is not qualified as to how it can be assumed

The reviewer is correct, we had not backed this statement up with data and it was more of an assumption. We removed this statement from the manuscript.

Line 233: Why weren't pH probes compared again here? The start of the section explains that they are

more difficult to use in PCR compared to LAMP, but it still needs to be shown how they actually compare to illustrate that the developed system is an improvement

The pH-based PCR was attempted in our previous manuscript establishing pH LAMP (Tanner et al Biotechniques 2015), but was extremely difficult as described in the text. The volume, mineral oil, and length requirements are directly from that work which we would just be duplicating for comparing to PAPS here. We feel the reference to previous work is sufficient.

Line 255: This concentration is high compared to the instrumental methods that are used for nucleic acid quantification. Standard absorbance measurements for example have detection limits in the low (less than 10) ng/uL and certain platforms (i.e. Nanodrop) are able to achieve sub ng/uL concentrations. While this approach cannot reasonably be compared to instrumental methods given that it is intended to be used in non-lab settings, it is also important to describe how these concentrations compare to existing techniques and methods (both those that are field-friendly and lab-based)

The DNA concentrations quantitated here refer to dsDNA PCR product yield after thermocycling measured using PicoGreen, so accordingly the values are high. We're only using the numbers to correlate yield and color change across the PCR cycles, with the PAPS signal potentially being useful as a DNA yield measurement, but the primary purpose is to estimate the yield required to cause color change not be a detection mechanism.

Reviewer #2 (Remarks to the Author):

Major Claims of the Paper-

This manuscript details a novel colorimetric amplification detection for LAMP utilizing pyridylazophenol dyes binding manganese ion, Mn²⁺. There is a need for a robust, low sophisticated instrumentation, pH independent, and strong visible color change in response to nucleic acid amplification within this loop-mediated isothermal amplification (LAMP) method to make this method more readily available. The author screened metal-sensing dyes for their ability to meet these goals. They found two related dyes, PAPS, that in the presence of Mn²⁺ produced a color change difference between a positive reaction and no template control reactions. This colorimetric detection worked robustly for both LAMP and PCR amplification. In addition, by taking the absorbance difference (A₄₅₀-A₅₅₀) and/or the ratio (A₄₅₀/A₅₅₀), a quantitative read-out accurately correlates with DNA yield.

The claims of this paper are novel and will be of interest to others in the application of LAMP and PCR methods. The colorimetric results are novel and will be of interest in the wider field of chemical biology.

Questions and concerns:

Does your assay require the opening of tubes to add the Mn²⁺? Is there a concern regarding contamination in your technique? According to the methods (line 88 and 89), "dyes and metal ions were diluted to 25x concentration and then mixed into LAMP reactions to 1x concentration before incubation",

what is the time between red to yellow color change in positive reactions?

This would indeed be a significant concern, but fortunately all the reactions are conducted in the presence of all components including PAPS and Mn. Color is visualized directly in the reaction vessel with no need to open the tubes or remove seals from plates. We would like this to be clear so we added some text to the methods clarifying the reaction setup. For the time to color change it follows DNA product accumulation and is most clearly seen with the PCR cycles, but in the LAMP reactions we generally incubate for 30-40 minutes but higher copy number reactions can give yellow color in 15-20 minutes by eye or faster if using more sensitive quantitative measurement such as absorbance. To further demonstrate this point we added Supplemental Figure 2 showing PAPS+Mn in two LAMP mixes, with clear visual and absorbance-measured color change occurring in ~20 minutes, paired with a fluorescence readout in about the same time demonstrating that the color change corresponds with DNA product accumulation. We also added text describing this at Line 195:

“We compared the speed of visual color change relative to the dsDNA accumulation during LAMP amplification with two commercial LAMP mixes (Supplementary Figure 2). The color change was also measured with a spectrometer and the difference between 450 nm and 550 nm was plotted. The results indicated that the visual color change is concurrent with the dsDNA accumulation, with visual and absorbance measurement appearing immediately following fluorescence signal from dsDNA intercalating dye, and color change visible by ~20 minutes incubation.”

What is the pH of your reactions before LAMP and after LAMP? I realize you are using a strong buffer but did you take the pH to be sure the reaction and/or addition of PAPS and metals did not exceed your buffer capacity? For example, the color of HNB changes depending on the pH of the solution. Does the PAPS color change depending on the color of solution? Ref 36 indicates that the PAPS chelate metal ions at optimum pH ranges.

The reaction contains 20 mM Tris at pH 8.8 and this is enough to keep the pH stable even with LAMP amplification. PAPS does display some pH dependence, but we found the red color for PAPS+Mn²⁺ remains when pH > 8.0 at room temperature. This is the same pH range of Tris-based buffers for most DNA polymerases when measured at room temperature and accordingly we did not have to modify the standard LAMP and PCR mixes we used. To clarify this we added a sentence in the Methods describing the LAMP condition: “LAMP reactions contained 20 mM Tris at pH 8.8 and standard concentrations of 1.4 mM each dNTPs and 8 mM MgSO₄.”

It would be interesting to see the pH dependency of Mn²⁺ with the PAPS. Ref 37 show 5-Br-PAPS chelating Co²⁺ at a particular pH (pH 7.0) with the addition of PDADMAC. This paper also suggests the structure of 5-Br-PAPS at pH 7 in the deprotonated phenol. In addition, Co²⁺ and 5-Br-PAPS in absence of PDADMAC caused a color change from yellow to red purple.

As noted there is certainly some pH dependence of the Mn-PAPS interaction and we saw poor color and color change below pH ~8, but did not pursue that further because the DNA amplification reaction also perform more poorly at close to neutral pH so this is a convenient

overlap. Perhaps Co or the other metals could work better at lower pH but the LAMP and PCR amplification will not. We did attempt to use PDADMAC for potential enhancement of metal-based color change but found it strongly inhibitory to LAMP and did not pursue it further, also it was clearly not necessary for Mn-dependent color change.

Any concerns about non-template amplification? Does the addition of pyridylazophenol dyes and manganese ion effect the LAMP reaction? Noticed the PAPS dyes and Mn²⁺ was added to the reactions prior to amplification which should prevent any contamination in the methods. But in the results, stated line 143 and 144 "when added to the amplification reaction mixture "is this after the amplification? If so are the tubes opened for the addition of Mn²⁺ after the amplification process while the 5-Br-PAPS is in the buffer solution before the amplification process?

All reaction components were added before the amplification just like in standard LAMP reaction set up, so the contamination risk remains the same as LAMP itself. We were cognizant of a potential increase in nonspecific amplification with the addition of Mn, but did not observe any and included NTCs with all experiments. To clarify these points we adjusted the text to read: "Mn²⁺ changed the color of 5-Bromo-PAPS from yellow to bright red when added in the reaction mix. Strikingly, after LAMP amplification, positive reactions reverted to the same yellow color as observed without Mn²⁺, while NTC reactions remained an unchanged red (Figure 1A, Supplementary Figure 1)"

Are the Pyridylazophenol dyes pH sensitive? Was the pH determined before and after the reaction to ensure there were no pH effects with this dye?

Yes to some degree, see discussion of pH in above comments.

Line 6 a comma is needed "but as the amplification techniques have moved away from the lab, complementary detection"

Line 11 manganese ion

Line 40 a comma and not a dash "large DNA yield-or the byproducts of its synthesis, can be exploited"

Line 53 no dash needed for "in a weakly-buffered solution"

Line 66 and 67 the N should be italicized in the name "2-(5-Bromo-2-pyridylazo)-5-[N-propyl-N-(3-sulfopropyl)amino]phenol" and "2-(5-Nitro-2-pyridylazo)-5-[N-n-propyl-N-(3-sulfopropyl)amino]phenol"

Line 96 within the Method: "elution buffer (10 mM Tris, pH8.5)" needs a space between pH 8.5

We thank the reviewer for noticing these errors, and have corrected them all in the revised manuscript.

Line 143 was Mn²⁺ added to the reaction mixture after the amplification process? Were the tubes opened in order to add the Mn²⁺? In the methods section, line 89 "concentration before incubation."

No, everything was present from reaction initiation as discussed and clarified above.

Line 146 add the word color "red color (Figure 1A, Supplementary Figure 1)." How much time did it take for the color change to occur?

How long does it take for the color to change from red to yellow in the positive reactions? For example reference 17 using calcein and Mn²⁺ the visualization takes 30-60 min.

Line 147 for the 5-Nitro-PAPS color change "back to yellow upon successful LAMP amplification" how much time did it take for this color change to occur?

See discussion and added data mentioned above for this point.

Line 155 add a comma and a transition such as "or other metal ions, and, thus accordingly focused"

Line 158 need to add 's to dye "thereby restoring the dye's original yellow color (Figure 2)."

Line 160 add (PPase) for "adding inorganic pyrophosphatase (PPase), which converts"

Line 163 spell out the chemical name for ethylenediaminetetraacetic acid (EDTA)

We thank the reviewer for noticing these errors, and have corrected them all in the revised manuscript.

Line 285 Figure 1 "lambda DNA (1ng)" needs a space between 1 ng. Why is the color different for the 100 μM and the 800 μM concentrations of Mn²⁺? Notice that there is no color change at the higher concentrations of Mn²⁺ (800 μM). The Real time graphs of RFU verses cycles are slightly blurry. Looks like the amplifications using the higher concentrations 800 μM Mn²⁺ didn't produce any RFU units until greater than 80 cycles. RFU y-axis is labeled in Figure 1E differently than the others.

We clarified this Figure description to read:

"In reactions with 800 μM Mn²⁺, the amplification was significantly impaired, as shown in the real time column of panels A, B and C. In addition, such limited amplification combined with the high concentration of Mn²⁺ could not produce a color change after 40 min incubation time (middle column in panels A, B), likely due to insufficient PPI."

The difference in visible dye color reduces the real time fluorescence emitted by the dsDNA binding dye, and thus the Y-axis for the real time curves is different due to automatic scaling by the software. We updated the Figures and final versions will be less blurry.

Line 294 Suggestion for Figure 2A to structurally follow the schematic 2C where the PAPS chelate the Mn²⁺; would the phenol be deprotonated and negatively charged like that in papers references 37 and 38? Did you take the pH of the samples before LAMP and after in the presence of Mn²⁺ to confirm there was no pH changes? Maybe interesting to see how the color of the PAPS change when chelating Mn²⁺ in a pH study similar to the referenced papers 37 and 38.

We thank the reviewer for this suggestion and changed the chemical structures to deprotonated forms. For simplicity, we kept a single form in 2A. The pH of reactions, which all contained 20 mM Tris, was stable before and after LAMP (in our experience with pH-dependent LAMP <5mM Tris is needed for large pH changes). In this work, we are most interested in the working condition for PAPS dye as an indicator for DNA amplification and how this dye changes color as a function of the pH is beyond the scope.

Line 300 Figure 3B why is there no data shown for 50 μM of Mn²⁺?

In this experiment, lambda template was inadvertently added to the NTC containing 50 μM of Mn²⁺, and thus removed from the image. To avoid confusion, those 6 reactions containing 50 μM of Mn²⁺ were removed altogether, since the data was not critical to any conclusions.

Line 167 In the paper state "Concentrations of each dye from 50-100 uM " however only Br-PAPS is shown in Figure 3.

The Nitro-PAPS dye was tested in the same manner as Br-PAPS, but was not significantly different and we preferred to focus on Br-PAPS for further experiments. This description has been modified to clarify this point: "Next, we determined optimal concentration ranges of 5-Bromo-PAPS and 5-Nitro-PAPS dyes and Mn²⁺ for colorimetric detection (shown for 5-Bromo-PAPS in Figure 3)."

Line 307 Figure 4. A, Absorption measured from 360 to 750 nm wavelength however your x-axis shows the measurement to 720 nm

This data was indeed a scan of 360-750 nm. We added tick marks to the updated graph and Figure to more clearly show the wavelength positions on the X-axis.

Suggestion: Change Mn⁺⁺ in figures with Mn²⁺ to match manuscript

We made this change and now write it as "Mn²⁺".

Line 307 Figure 4. D, also with concentrations with 75 uM 5-Bromo-PAPS and 75-150 uM Mn²⁺ should be

added to legend. What is the 1ng-1B, 1ng-2B, and 1ng-3B in Figure 4? Assuming reactions in triplet?

We added these descriptions and clarifications to the legend, with the reactions indeed representing replicate measurements.

Table 1 suggestion change Mn⁺⁺ to Mn²⁺ and define N2, change +GuCl to GuHCl to be consistent, define E1 and E1700.

This table has been adjusted as suggested, and N2 and E1 defined in the text. We arranged the test conditions for clearer presentation and removed the commercial name E1700 which is for the base condition without PAPS dye or Mn²⁺.

Line 209 add a space "Qiagen EB, 10mM Tris pH 8.5)" to 10 mM

This change was made in the text.

Line 215 "Tolerance in quantitative measurement systems, e.g., absorbance, is likely" remove the commas and add parentheses, measurement systems (e.g. absorbance) is likely
As noted above this sentence was removed from the text.

Line 226 and 227 no dash for the no template control reactions or could use NTC

We changed these to "NTC".

Line 231 What are the standard PCR mixes (e.g. and list them)? Are these the three chosen in Figure 6A?
Yes, this was indicating the same three commercial PCR mixes in Figure 6. We updated this text to read "generally compatible with commercial PCR mixes."

Line 276 and 277 perhaps you should also include manganese ion "the use of PAPS dyes and manganese ion" overcome

Line 338 Figure 7. "the annealing temperature is shown" should include °C perhaps in parenthesis after temperature (°C)

Line 339 B "bacterial cells using primes recognizing plasmid I." should be primers?

Line 339 C "bacterial cells using primes recognizing plasmid II." Should be primers?

Line 340 D "bacterial cells using primes recognizing both plasmids" Should be primers?

We thank the reviewer for these suggestions and clarifications, all changes were made in the text.

Suppl. Figure 1B and 1C

Line 2 the chemical formulas should include subscripts. Did you mean to use iron(II) sulfate and not chloride?

Yes, this is correct, and we made this change in the Supplement.

Suppl. Figure 2:

Line 12 In the legend "visual detection is shown for 5-Bromo-PAPS (PAPS)" add (PAPS) to be consistent with images. Also replace Mn⁺⁺ with Mn²⁺

This has been changed as suggested.

Suppl Figure 3:

Line 22 Space between "1ng lambda DNA" to 1 ng

Line 23 Space between "0.5kb amplicon" to 0.5 kb

Line 24 and 25 Space between "450nm and 550nm" to 450 nm and 550 nm

In A for both Suppl Figure 3 and Suppl Figure 4 should also have spaces between 0.5 kb, 1.0 kb, 2.0 kb

All of these changes have been made on the Figure.

References

Reference 5 line 354, reference 7 line 359, reference 12 line 372, reference 23 line 402, reference 25 line 407, reference 33 line 430 remove et al and include all the authors

Reference 37 line 429 and reference 39 line 445 superscript Cu²⁺

We reformatted all the citations and references to Nature style and corrected these errors.

Reviewer #3 (Remarks to the Author):

Recommendation: Accept with major changes.

This manuscript describes an indicator for detection of amplification of DNA by PCR and RT-LAMP methods. The test detects the hydrolysis of the nucleotide triphosphates concomitant with DNA polymerization. The idea is to carry out the amplification in the presence of 50 micromolar Mn(II) and 50 micromolar Mn(II) indicator dye. Manganese(II) binds strongly to the inorganic pyrophosphate (PPi) formed during DNA polymerization, so when the concentration of the ds amplicon rises into the micromolar range, enough PPi is formed to pull the Mn(II) off the dye, and the dye changes color. The work is solid, the materials and methods are described adequately, and its immediate applicability to NAT visualization makes it suitable for the readership of Comm. Bio. I'd be interested in giving it a go.

→This method provides a simple visual mechanism for detection of a NAT test result.

→The method is superior to the pH indicator dyes that are used in commercial RT-LAMP COVID tests because it tolerates carryover of buffers and can be used on matrices with a wide range of pH values.

The authors screened several Mn(II) indicator dyes and found one with a strong color change (red to yellow). The suggested mechanism for the color change was tested by adding an enzyme to hydrolyze PPI to phosphate (which does not bind Mn(II) as strongly) and by using EDTA to chelate the Mn(II). They tested the method for detection of RT-LAMP amplification of SARS-COV2 RNA, finding performance comparable to that of RT-LAMP with pH indicator dyes, and higher tolerance to carryover from sample prep stages. The mss claims that the dye and Mn(II) had no impact on the detection sensitivity, I think a little more work is needed to establish this (see change #1).

They then set up a variety of PCR amplifications (lambda DNA of several lengths, E. coli and human DNA fragments) using different polymerases and monitored the DNA production with a conventional fluorescent dsDNA dye. The utility of the dye test for monitoring plasmid transfection in place of gel electrophoresis is nicely shown in Figure 7.

Major changes:

1) The data in table 1 show positives and negatives detected by a conventional DNA fluorescent dye from 24 samples at analyte concentrations near the LOD for the RT-LAMP test; results range from 4/24 to 21/24 positives. It's hard to see which differences are significant and which are due to random error. It would strengthen this data to show a replication and some statistical analysis.

As described above we are using this assay below its LOD to observe any potential impact from the Mn and PAPS additions, with the effects of combining primer sets and guanidine hydrochloride demonstrated in previous more extensive studies of the assay. A good amount of the spread in results is indeed from the different sensitivity when using 1 vs. 2 primer sets, adding in Gu HCl, etc. but we wanted to examine any Mn/PAPS impact with all conditions. To address the comment here though we added in a Two proportion Z-test (two-tailed) for the control condition vs. the Mn or PAPS addition; in almost all conditions the difference is likely to be due to chance as indicated by the high p-values. However we measure a p-value <0.05 in the dual-primer+Gu HCl conditions when adding PAPS or Mn; this indicates a potential positive impact of adding Mn or PAPS which we now describe in the text, but do not claim to be proven, only our original intent to determine if there were any *negative* effect from adding Mn or PAPS.

Minor changes:

2) The optimal concentrations of the dye and Mn(II) are different for LAMP and for PCR, please add some explanation in the text.

"For subsequent studies we chose 75 μ M PAPS dye with 100 μ M Mn²⁺ for optimal visual contrast." In materials and methods, "Colorimetric detection of PCR amplification was performed in 25 μ L reactions by including 50 μ M PAPS dye and 50 μ M Mn²⁺ into reactions"

The reviewer is correct, the apparent best conditions were slightly different for PCR and LAMP, likely due to the very different formulations: the PCR mixes contain 1.5-3 mM Mg and 0.2 each mM dNTP, the LAMP mixes 8 mM Mg and 1.4 mM each dNTP (and accordingly make much more DNA product). We added some text to this effect, with the manuscript now stating:

“We tested various concentrations of 5-Bromo-PAPS and Mn^{2+} and found that each at 50 μ M produced the best visual contrast in the PCR mixes. This is slightly different from the best conditions for LAMP, likely due to the significant differences in Mg^{2+} , dNTP, and DNA yield in the different reactions.”

3) typos

a) line 200 and 203 use *GnHCl* to abbreviate *guanidinium HCl*.

b) Figure 7 caption says *primes* instead of *primers*.

We thank the reviewer for noticing these errors and have corrected them in the text.

REVIEWERS' COMMENTS:

Reviewer #1 (Remarks to the Author):

The authors addressed all my concerns with the manuscript and I have no further comments or questions.

Reviewer #2 (Remarks to the Author):

Reviewer #2-Major Claims of the Paper

(Comments also attached)

This manuscript details a novel colorimetric amplification detection for LAMP utilizing pyridylazophenol dyes binding manganese ion, Mn^{2+} . There is a need for a robust, low sophisticated instrumentation, pH independent, and strong visible color change in response to nucleic acid amplification within this loop-mediated isothermal amplification (LAMP) method to make this method more readily available. The author screened metal-sensing dyes for their ability to meet these goals. They found two related dyes, PAPS, that in the presence of Mn^{2+} produced a color change difference between a positive reaction and no template control reactions. This colorimetric detection worked robustly for both LAMP and PCR amplification. In addition, by taking the absorbance difference (A450-A550) and/or the ratio (A450/A550), a quantitative read-out accurately correlates with DNA yield.

The claims of this paper are novel and will be of interest to others in the application of LAMP and PCR methods. The colorimetric results are novel and will be of interest in the wider field of chemical biology.

I feel as if my concerns were adequately addressed. There were, however, a few typos that could be corrected to make the paper better quality for publication. These include the following:

Typo 1: The author's response:

"The reaction contains 20 mM Tris at pH 8.8 and this is enough to keep the pH stable even with LAMP amplification. PAPS does display some pH dependence, but we found the red color for PAPS+ Mn^{2+} remains when pH > 8.0 at room temperature. This is the same pH range of Trisbased buffers for most DNA polymerases when measured at room temperature and accordingly we did not have to modify the standard LAMP and PCR mixes we used. To clarify this we added a sentence in the Methods describing the LAMP condition: "LAMP reactions contained 20 mM Tris at pH 8.8 and standard concentrations of 1.4 mM each dNTPs and 8 mM $MgSO_4$."

Typo: The 4 needs to be subscripted in the text for Magnesium sulfate.

Typo 2: The author's response:

"We thank the reviewer for this suggestion and changed the chemical structures to deprotonated forms. For simplicity, we kept a single form in 2A."

Figure 2A: The chemical structures were changed to show the deprotonated forms but the alignment on the drawing program for the "ONa" and "Br" covalent bond is not correctly aligned. This could be improved for publication quality.

Typo 3:

Revision Figure 4. line 346 space between 1 and ng "in triplicate with 1 ng of lambda DNA (1ng-1B, .." and in Figure 4c. needs to have the -0.32 contained within the graph

Typo 4: Reviewer: Line 209 add a space "Qiagen EB, 10mM Tris pH 8.5)" to 10 mM

Author's response: This change was made in the text.

However, Line 233 change still needs to be made.

Typo 5: Reviewer: Suppl. Figure 1B and 1C Line 2 the chemical formulas should include subscripts. Did you mean to use iron(II) sulfate and not chloride?

Author's response: "Yes, this is correct, and we made this change in the Supplement."

If this is true, then iron(II) sulfate needs to be changed to iron(II) chloride in the Suppl Figure 1b and 1c and also in the methods. No changes appear to have been made. Also, all the chemical formulas need to be subscripted within the image.

Typo 6: Author's response: "We reformatted all the citations and references to Nature style and corrected these errors."

Does Nature allow et al in the references or do all the names associated with the publication need to be listed within the references?

Is "et al." acceptable in the References? Reference numbers 5, 7, 8, 23, 25, 32, 33, 40, 41, 43

Reviewer #3 (Remarks to the Author):

The main issue I was doubtful about was the significance of the results from the very low copy number amplification run (which also raised some questions from the other reviewers). The statistical analysis added to the mss shows the reader which of the outcomes were significant. This revision of the manuscript addressed my concerns, and I think it is ready for publication.